# Evaluation of African Swine Fever Virus *E111R* Gene on Viral Replication and Porcine Virulence

**DOI:** 10.3390/v15040890

**Published:** 2023-03-30

**Authors:** Xintao Zhou, Jiaqi Fan, Yanyan Zhang, Jinjin Yang, Rongnian Zhu, Huixian Yue, Yu Qi, Qixuan Li, Yu Wang, Teng Chen, Shoufeng Zhang, Rongliang Hu

**Affiliations:** 1College of Life Sciences, Ningxia University, Yinchuan 750021, China; 2Key Laboratory of Prevention & Control for African Swine Fever and Other Major Pig Diseases, Ministry of Agriculture and Rural Affairs, Changchun 130122, China; 3Changchun Veterinary Research Institute, Chinese Academy of Agricultural Sciences, Changchun 130122, China

**Keywords:** African swine fever, African swine fever virus, *E111R* genes, virulence, dose-dependent

## Abstract

African swine fever (ASF) is an acute infectious disease of domestic pigs and wild boars caused by the African swine fever virus (ASFV), with up to a 100% case fatality rate. The development of a vaccine for ASFV is hampered by the fact that the function of many genes in the ASFV genome still needs to be discovered. In this study, the previously unreported *E111R* gene was analyzed and identified as an early-expressed gene that is highly conserved across the different genotypes of ASFV. To further explore the function of the *E111R* gene, a recombinant strain, SY18ΔE111R, was constructed by deleting the *E111R* gene of the lethal ASFV SY18 strain. In vitro, the replication kinetics of SY18ΔE111R with deletion of the *E111R* gene were consistent with those of the parental strain. In vivo, high-dose SY18ΔE111R (10^5.0^ TCID_50_), administered intramuscularly to pigs, caused the same clinical signs and viremia as the parental strain (10^2.0^ TCID_50_), with all pigs dying on days 8–11. After being infected with a low dose of SY18ΔE111R (10^2.0^ TCID_50_) intramuscularly, pigs showed a later onset of disease and 60% mortality, changing from acute to subacute infection. In summary, deletion of the *E111R* gene has a negligible effect on the lethality of ASFV and does not affect the viruses’ ability to replicate, suggesting that *E111R* could not be the priority target of ASFV live-attenuated vaccine candidates.

## 1. Introduction

African swine fever (ASF) is an acute, hemorrhagic, febrile infection with manifestations including fever, cough, anorexia, diffuse epidermal hemorrhage, and other non-specific clinical signs; a markedly enlarged spleen is usually visible on autopsy [1]. African swine fever virus (ASFV) is the causative agent of ASF and belongs to the genus *Asfivirus* of the family *Asfarviridae* [2]. ASFV can infect all species of pigs and causes up to 100% mortality in domestic and wild boars, with each epidemic posing a serious challenge to the pig industry in infected areas [3,4]. ASF was first identified in Kenya in eastern Africa in the 1920s, and for a long time, ASFV was only endemic south of the Sahara Desert on the continent [5]. In the mid-20th century, the ASFV genotype I virus spread outwards from Africa to Europe and South America via international flights. In the 1990s, due to strict control, the disease was eradicated everywhere other than in Africa and Sardinia (Italy) [6,7]. However, a new global ASFV epidemic emerged in the 21st century, with the ASFV genotype II virus spreading to Georgia in the Caucasus region in 2007 [8]. The ASFV genotype II virus spread westward to Eastern and Central Europe in 2014 and eastward to East and Southeast Asia in 2018, where the virus is still spreading [9,10,11,12,13]. In 2021, genotype II ASFV re-emerged in the Dominican Republic and Haiti in the Caribbean after a gap of 40 years [14].

The continued spread of the virus makes it increasingly difficult to prevent ASF outbreaks in countries, currently free from ASFV. Vaccines are the best option for ASFV prevention, although previous findings have shown that inactivated ASFV vaccines do not work [15]; gently engineered vaccines are unstable in their protective properties; and some naturally attenuated and modified ASFVs are immunoprotective but can cause side effects, and thus need to be used with caution [16,17,18,19]. The artificial gene-deficient ASFV weak strain vaccine offers better protection, but given its biosafety concerns, no vaccine has been approved for marketing in most countries and regions except Vietnam [13].

The ASFV genome is between 170 and 194 kb in length and encodes 151–174 open reading frames; only a tiny proportion of these genes have been studied for their function [20,21]. Researchers have found that the deletion of some genes from the virus attenuates ASFV virulence and does not cause clinical signs of ASF in domestic pigs. Among them, *9GL(B119L)*, *UK(DP96R)*, *CD2v(EP402R)*, *MGF505-1R*, *MGF505-2R*, *MGF505-3R*, *MGF360-12R*, *MGF360-13R*, *MGF360-14R*, *DP148R*, *I177L*, *I226R*, and other genes were deleted to attenuate the virulence of ASFV [18,19,22,23,24,25]. The I177L and I226R genes are located at the right-hand end of the central constant region (CCR) of the ASFV genome, and their deletion allows ASFV to be completely attenuated and retain good immunogenicity [26,27]. The E120R gene and I329L in the same region affect the immunosuppression or virulence of the virus [28,29].

The *E111R* gene, an early-expressed gene in ASFV, is positioned at the right-hand end of the CRR of the ASFV genome and has 111 amino acids; there are no genes in the gene database which are identical to the *E111R* gene. There has yet to be any research concentrating on the *E111R* gene; therefore, we aim to determine whether the *E111R* gene affects replication and virulence in ASFV. Through an in vitro homologous recombination method, the *E111R* gene of the lethal ASFV SY18 strain was replaced by a green fluorescent protein (EGFP) expression cassette in a recombinant plasmid, yielding a recombinant strain with the *E111R* gene deleted (SY18ΔE111R). The ability of SY18ΔE111R to replicate in vitro and its virulence in pigs were also tested. Piglets were infected intramuscularly with a high dose (10^5.0^ TCID_50_) of SY18ΔE111R, and all animals showed typical clinical signs of acute ASF. Inoculating piglets with a low dose (10^2.0^ TCID_50_) of SY18ΔE111R resulted in a 40% survival rate and reduced clinical signs. The results show that deletion of the *E111R* gene has no effect on ASFV SY18 in vitro replication and does not significantly diminish the virulence of the strain.

## 2. Materials and Methods

### 2.1. Viruses and Cells

The ASFV SY18 strain (GenBank number: MH766894.2) is a highly virulent strain of genotype II ASFV that causes acute infection and 100% mortality in pigs isolated from tissues of ASF-dead pigs collected in Shenyang, China, in 2018 [10,18]. The ASFV SY18 used in this experiment is a virus that has been passaged six times consecutively on primary bone marrow macrophages (BMDMs) and stored at −80 °C.

In this study, we used BMDMs and primary alveolar macrophages (PAMs) isolated from healthy piglets of around 2 months of age, and the cells were prepared as reported in the literature [18,30,31]. BMDMs were stimulated with 10 ng/mL granulocyte–macrophage aggregate (GM-CSF) (a porcine GM-CSF protein derived from *E. coli*, sequence reference from Gene ID: 397208 in the National Center for Biotechnology Information (NCBI), prepared in our laboratory) for 7 to 10 days to allow cell differentiation to enhance cell infectivity [31]. Exogenous testing was performed in the PAMs and BMDMs to exclude contamination, with reference to the methods already described [32]. The following pathogens of swine diseases were tested: ASFV, pseudorabies virus (PRV), classical swine fever virus (CSFV), porcine reproductive and respiratory syndrome virus (PRRSV), porcine circovirus 1/2 (PCV1/2), and porcine parvovirus (PPV).

### 2.2. Amino Acid Sequence Analysis

The homology of the E111R gene at the amino acid level was analyzed between the ASFV SY18 strain and different genotypes of ASFV isolates. The amino acid sequences were compared with multiple alignments using fast Fourier transform (MAFFT) (https://www.ebi.ac.uk/Tools/msa/mafft/, accessed on 20 November 2022) and Jalview 2.11.2.0 software (https://www.jalview.org/, accessed on 20 November 2022). The dataset submitted in MAFFT (supplementary submission) is available at http://www.ebi.ac.uk/Tools/services/rest/mafft/result/mafft-I20230206-020914-0491-53319287-p1m/aln-fasta, accessed on 20 November 2022.

### 2.3. E111R Gene Expression Characteristics

PAMs were seeded in a 12-well plate (Corning, Wujiang, China) and cultured overnight. Then, ASFV SY18 (MOI = 3), inhibitor groups (100 ug/mL inhibitor cytosine arabinoside (AraC) + ASFV SY18 (MOI = 3)) and equal amounts of cell cultures were added for control, and cultures were collected at 1, 4, 8, 12, 16, and 24 hours post-inoculation (hpi) and stored at −80 °C [33]. Three replicates were set at each time. All cultures were melted together on the ice and handled in a uniform manner. Total cellular ribonucleic acid (RNA) was extracted using the FastPure Cell/Tissue Total RNA Isolation Kit V2 (Vazyme, Nanjing, China). Complementary deoxyribonucleic acid (cDNA) was synthesized by reverse transcription according to the HiScript III RT SuperMix for polymerase chain reaction (qPCR) (Vazyme) instructions, and real-time fluorescent qPCR was performed according to the ChamQ Universal SYBR qPCR Master Mix (Vazyme) instructions. The 2^−ΔΔCq^ method was used to analyze the expression of target genes, with glyceraldehyde-3-phosphate dehydrogenase (*GAPDH*) used as an internal reference gene [34]. The designed qPCR primers are shown in Table 1.

### 2.4. Construction of Recombinant ASFV SY18∆E111R

The recombinant virus without the E111R gene was created using the homologous recombination between the parental SY18 genome and the recombinant transfer vector [18]. The recombinant plasmid (p-p72EGFP∆E111R) contained a 1125 bp homologous arm (167,981–169,105 bp) on the left flank of the E111R gene, a p72 promoter, and a reporter cassette for the EGFP gene (p72-EGFP-SV_40_ polyA) containing sequences on both sides of the E111R gene and the EGFP gene; and a 1175 bp homologous arm on the right flank of the E111R gene (169,418–170,602 bp) (Figure 1a). The p72 promoter cloned from the ASFV genome initiated the expression of the EGFP gene. After sequencing to determine that the inserted homologous arms and expression cassette sequences were free of errors, the recombinant plasmid was transfected into BMDMs using the jetPEI^®^-Macrophage DNA transfection kit (Polyplus, Strasbourg, France). Four hours after transfection, cells were infected with ASFV SY18 so that the genome of SY18 and the recombinant plasmid could undergo homologous recombination within the cells. At 24 hpi, cells with green fluorescent protein were observed and screened under a fluorescent microscope (Olmpus-IX73, Tokyo, Japan). A mixture of the mutant virus and the parental strain was obtained at this point. This virus mixture was then seeded in the PAM after a 10-fold gradient dilution to exclude the parental virus with green fluorescence. The previous step was repeated, and the final purified mutant virus, named SY18ΔE111R, was obtained. Subsequently, SY18ΔE111R was amplified on BMDMs and passaged twice, named 1st generation SY18ΔE111R and 2nd generation SY18ΔE111R, respectively, for use in subsequent experiments.

The purity of SY18ΔE111R was identified by a polymerase chain reaction (PCR). If the parental strain ASFV SY18 was present, a 284 bp fragment was amplified by the forward primer (5’-TTAGCGAATGTCCCTTAGTT-3’) and reverse primer (5’-CGTACAGTCCTTCCAGTTAT-3’). Meanwhile, to verify that the SY18ΔE111R genome did not contain other mutations, DNA from the 2nd generation SY18ΔE111R was next-generation sequenced on an Illumia novaseq6000, PE150 (Novogene Ltd., Tianjin, China).

### 2.5. SY18ΔE111R Growth Curve

PAMs were spread in a monolayer on a 12-well plate (Corning, Wujiang, China) and incubated for 12 h. The PAMs were infected with ASFV SY18, and SY18ΔE111R at a concentration of 0.1 MOI, and cultures were collected at 2, 6, 12, 24, 48, 72, 96, and 120 h. Three replicates were set at each time. Collected cell cultures were freeze–thawed three times for a median titration of tissue culture infectious dose (TCID_50_)/mL. The PAMs were then inoculated in 96-well plates (Corning, Wujiang, China), and viral cultures were inoculated onto cells using a 10-fold gradient dilution and incubated for 5 days. Cell cultures infected with SY18 were analyzed by immunofluorescence using the monoclonal antibody against the p30 protein labeled by fluorescein isothiocyanate (FITC) (prepared in our laboratory). Cells infected with SY18ΔE111R could also be viewed directly under a fluorescent microscope. Viral titers were calculated using the Reed–Muench method [35].

### 2.6. Animal Experiments

Animal experiments were conducted in an Animal Biosafety Level 3 (ABSL-3) laboratory. Fifteen experimental piglets (landrace and Yorkshire crosses, 2–3 months old, weighing approximately 15 kg) were purchased for this experiment. Blood from the pigs was tested for the absence of common swine disease viruses (ASFV, CSFV, PRRSV, PRV, PPV, and PCV1/2) using the same exogenous assay as for the PAMs and BMDMs in Section 2.1, and the pigs were negative for antibodies for p54 in their serum [32].

Animal experiments include three groups. The first group was a control group named SY18, consisting of five pigs (No. SY18-1 to SY18-5) each injected intramuscularly with a low dose (10^2.0^ TCID_50_) of ASFV SY18. The second group was the EL group (*n* = 5), consisting of five pigs (No. EL-1 to EL-5), each injected intramuscularly with a low dose (10^2.0^ TCID_50_) of the 2nd generation SY18ΔE111R. The third group was the EH group, consisting of five pigs (No.EH-1 to EH-5), each injected intramuscularly with a high dose (10^5.0^ TCID_50_) of the 2nd generation SY18ΔE111R. The SY18ΔE111R used was titrated according to TCID_50_/mL. To avoid errors, the uniform inoculation site is the buttocks, and the uniform inoculation volume is 1 mL.

Observe and record clinical signs (rectal temperature, fever, anorexia, depression, diarrhea, staggering feet, trembling, coughing, purple skin, and prolonged recumbency) in domestic pigs with reference to King et al.’s ASF clinical score [36]. Oral swabs, anal swabs, and blood were collected from animals at 7-day intervals. For ASFV nucleic acid detection, 500 μL of the peripheral blood sample was collected in EDTA-containing tubes, and the serum was isolated using other normal blood. During observation, pigs showing severe clinical signs were euthanized with pentobarbital. Tissues (submaxillary lymph nodes, heart, lung, thymus, bone marrow, liver, spleen, kidney, colon, inguinal lymph nodes, and joint fluid) from euthanized pigs are collected and assayed by RT-qPCR to quantify the ASFV nucleic acid in them.

### 2.7. Virus Detection in Blood and Tissue

The viral load in the blood and tissues of animals was measured by using real-time fluorescent quantitative PCR probes targeting the ASFV B646L (p72) gene. The primer synthesis and reaction conditions are recommended by the National Standards of the People’s Republic of China (GB/T 18648-2020) (https://openstd.samr.gov.cn/bzgk/gb/index, accessed on 20 November 2022), as shown in Appendix A. A standard curve check was also established based on a standard plasmid (developed in our laboratory by inserting the B646L gene sequence at the *Xba* I site of the pUC19 plasmid) for the ASFV SY18 B646L gene and the copy number of the ASFV genomic samples was calculated.

### 2.8. Detection of Anti-p54 Antibodies

ELISA plates were coated with 1 µg/mL purified p54 protein (reference sequence of GeneID:59226978, prepared in our laboratory using *E. coli*, specific for ASFV antibody-positive sera) as described previously [18], and the expression of the antibody to the ASFV-specific p54 was detected using an indirect ELISA, which was repeated three times for serum samples collected at 0, 7, 14, and 21 days post inoculation (dpi). The absorbance of the sample was measured at 450 nm as the sample (S) value, and the ASFV rehabilitation pig serum was measured as the positive (*p*) value. The cut-off value was 0.25 and samples were considered positive when S/P > 0.25 and negative when S/P ≤ 0.25.

### 2.9. Statistical Analysis

Figures were plotted using GraphPad Prism 9.4.1 software (https://www.graphpad.com/, accessed on 20 November 2022), and data on growth characteristics were statistically analyzed using the Holm–Šidák test therein [37]. The *p*-value < 0.05 was considered statistically different, and the *p*-value> 0.05 was considered statistically insignificant.

## 3. Results

### 3.1. Conserved Nature of the E111R Gene in Different Isolates

The open reading frame (ORF) *E111R* of ASFV SY18 strain consists of 336 bp and encodes a protein of 111 amino acids. It is located on the SY18 genome between nucleotides 169,105 and 169,440 on the positive strand. Currently, no genes similar to the E111R gene (except ASFV) were searched in the NCBI gene database using the Basic Local Alignment Search Tool (BLAST). There are no strains of ASFV that exhibit deletion of the *E111R* gene in the database. The *E111R* amino acid sequences of the different genotypes of ASFV were analyzed using the MAFFT online website and Jalview software by downloading the ASFV gene sequences of the different genotypes on NCBI. The results showed that in all 16 ASFV strains compared, the 111 amino acid homology encoded by the E111R gene ranged between 95.5% and 100% and was more conserved in ASFV strains of genotype I and II (Figure 2).

### 3.2. The E111R Gene Is an Early Expressed Gene in ASFV Infection

Based on the specific accumulation kinetics of ASFV, its gene expression can be divided into four phases, separated by DNA replication. Those expressed before DNA replication are known as pre-replication (or exceedingly early) and early genes; those expressed after DNA replication has begun are termed intermediate and late genes [19]. It has been documented that the replication life cycle of ASFVs is 18 hpi, with the *B646L* (p72) gene being a late-expressed gene and the *CP204L* (p30) gene being an early-expressed gene. The results showed that with *GAPDH* as the housekeeping gene, the *CP204L* (p30) gene reached a plateau at 4 hpi, while the expression of the *B646L* (p72) gene continued to rise (Figure 3). The *E111R* gene increased rapidly from 0 to 4 hpi after infection and reached a plateau at 8 hpi. To further differentiate and validate the transcriptional kinetics of the E111R gene in the ASFV genome, we also examined the expression of p30, p72, and *E111R* genes under the presence of AraC inhibitors. At the beginning of 8 h, AraC blocked the transcription of the p72 gene but not the p30 and *E111R* genes. This demonstrates that transcription of both *E111R* and p30 occurs early in the infection and that *E111R* is an early transcribed gene in ASFV. This result complements the data from the ASFV transcriptome article published by Cackett et al. in 2020 [38].

### 3.3. Purification and In Vitro Growth Characteristics of the SY18ΔE111R Strain

To determine the function of the *E111R* gene, we constructed an *E111R* gene deletion recombinant virus (SY18ΔE111R) on a highly virulent strain prevalent in China [10,18]. Most researchers have successfully built ASFV gene deletion strains by homologous recombination. As shown in the experimental steps in Figure 1a, the entire 336 bp base will be missing because the ORF of the *E111R* gene does not share a nucleobase with the ORFs of the left and right *EP296R* and *E66L* genes. We used the EGFP gene (green fluorescent) instead of the *E111R* gene. Through 15 rounds of limited dilution, a purified recombinant virus was obtained, which was named SY18ΔE111R. Subsequently, the 1st and 2nd generation SY18ΔE111Rs for use in subsequent experiments were obtained by amplifying SY18ΔE111R twice passaged on BMDMs.

By PCR identification, the 1st and 2nd generation SY18ΔE111R viruses did not amplify a 248 bp fragment (Figure 1b). The next-generation sequencing results showed that the sequence variation of the 2nd generation SY18ΔE111R was as expected. Figure 1c shows the growth status of 2nd generation SY18ΔE111R after 96 hpi with PAMs. The replication kinetics of the 2nd generation SY18ΔE111R were similar to those of SY18 in vitro (Figure 1d), with no significant difference in growth rate between the two (*p* > 0.05) and no significant difference in the titer between the two strains at 120 h (*p* > 0.05). This demonstrated that deletion of the *E111R* gene had no significant effect on the replication of the SY18ΔE111R strain in vitro compared to the parental wild strain ASFV SY18. The E111R gene is not a replication-associated gene in the ASFV genome.

### 3.4. Virulence of SY18ΔE111R in Pigs

To assess the effect of the deletion of the E111R gene on ASFV virulence, we infected pigs with the 2nd generation SY18ΔE111R strain with the deletion of the *E111R* gene by intramuscular injection at a dose of 10^2.0^ TCID_50_ per head (*n* = 5) and 10^5.0^ TCID_50_ per head (*n* = 5), while setting up a control group (*n* = 5) with intramuscular injection of the ASFV SY18 strain at a dose of 10^2.0^ TCID_50_ per head. During the post-inoculation observation period, it was found that all three groups of pigs recorded clinical signs of fever, anorexia, and diarrhea. Pigs infected with high doses of SY18∆E111R (10^5.0^ TCID_50_) showed similar clinical signs to those infected with low doses of ASFV SY18 (10^2.0^ TCID_50_). All showed clinical signs associated with ASF (fever, anorexia, diarrhea, purple skin, staggering feet, and trembling), mortality was 100%, and days to survival, time of initial fever, duration of fever days, and maximum fever temperature did not differ significantly (Table 2, Figure 4). During the 21-day observation period, two of the five pigs infected with a low dose of SY18∆E111R (10^2.0^ TCID_50_) survived with a 60% mortality rate, delayed initial fever, and death, and milder clinical signs (e.g., fever and diarrhea) (Table 2, Figure 4a). Rectal temperature changes for all pigs are shown in Figure 4b.

We collected oral swabs, anal swabs, and blood samples from pigs in the experimental group on days 0, 7, 14, and 21 to quantify the ASFV genomic DNA in them, and the results are summarised in Table 3. Both pigs infected with the high dose of SY18∆E111R and pigs infected with ASFV SY18 developed hyperplasma viremia on the 7th day, and all died by the 10th day of immunization. In contrast, pigs in the EL group developed viremia and morbidity much later after intramuscular injection of SY18ΔE111R at 10^2.0^ TCID_50_ (low dose). Of these, EL-2 pigs developed detectable viremia at 7 days post-inoculation (dpi) and were subsequently euthanized at 12 dpi; EL-1 and EL-4 pigs developed detectable viremia at 14 dpi and were euthanized at 15 and 17 dpi; EL-3 and EL-5 pigs, which survived to the end, did not develop detectable viremia until 21 dpi.

All pigs exhibited detectable virus in oral swabs at 7 dpi (except for the EH-3 pig). Anal swabs were more confusing but were still able to identify pigs infected with low doses of SY18∆E111R and surviving to day 21 with late detection of the virus in the anal swab.

Pigs in the low-dose SY18∆E111R group produced antibodies to p54 (EL-2 pig, the earliest to die, was weakly positive for p54 on day 7); however, pigs in the high-dose SY18∆E111R and ASFV SY18 groups were consistently negative for p54 (Figure 4c).

Tissues from euthanized pigs were also quantified for ASFV nucleic acid in them by the RT-PCR method. As with the swab results, all individuals (including EL-3 and EL-5 pigs that survived to day 21 after infection) had positive tissues. In particular, the spleen, submandibular lymph nodes, and bone marrow had high levels of viral load (10^5.0^–10^8.5^ copies/mL) (Figure 5).

## 4. Discussion

With international trade, the spread of the ASFV has become more rapid. In 2018, China reported an outbreak of ASF, followed by 14 surrounding countries reporting the disease within a year [38,39,40,41,42]. Nowadays, there is no cure, and the vaccine is in the laboratory stage, so strict quarantine, prevention, and control measures are the only way to stop the spread of ASFV. China has a vast territory and a large-scale pig industry, producing the most pork worldwide. The prolonged spread of ASF in China has caused huge economic losses, making a vaccine an urgent need.

Currently, reported ASFV isolates vary in genome length, with the longest length of the genome being 193,886 bp (Kenya_1950, type Ⅹ, AY261360.1) and the shortest length of the genome being 170,101 bp (BA71V, type I, U18466.2), the main difference being the left and right variable regions of the genome, but both encoding over 150 ORFs [21,43,44,45]. However, about half of the genes in the ASFV genome are currently unspecified as to whether they are associated with the virulence of the virus, which poses a continuing difficulty for vaccine development. Initially, genes in the left and right variable or central regions of the genome, such as CD2v (EP402R), UK (DP96R), 9GL (B119L), and MGFs, were selected for the construction of gene deletion strains [23,46]. Recently, the right-hand end of the CCR of the ASFV genome has become a key region for research. Of these, the I177L and I226R genes are the best gene deletion vaccine candidates identified to date, with a single deletion allowing complete attenuation of highly virulent ASFV strains and resistance to muscle challenge by high doses of parental strains [18,19,47]. In addition, the EP296R gene and the I329L gene in this region have also been reported to be associated with virulence [28,48].

The *E111R* gene is a gene of unknown function at the 3’ end of the CCR of the ASFV genome, and there are no genes or proteins similar to it in the database. *E111R* is highly conserved in ASFV isolates of different genotypes, with 95.5% to 100% amino acid homology, and the amino acid sequence of *E111R* is identical in several genotype I and genotype II strains. The recombinant virus SY18∆E111R has a similar replication capacity to ASFV SY18, demonstrating that the *E111R* gene is not essential for viral replication, as are the *EP296R*, *I329L*, and *I267L* genes, also located at the right end of the CCR [28,31,48].

In animals, different doses of SY18∆E111R have shown different survival rates and clinical signs. The high-dose SY18∆E111R group and the ASFV SY18 group showed the same mortality rate as the acute infection, causing similar clinical symptoms. The low-dose SY18∆E111R-infected pigs became sub-acutely infected with a survival rate of 40%. This demonstrates that the *E111R* gene affects a small proportion of the virulence in ASFV, causing clinical signs in line with the doses injected. Pigs in the low-dose SY18∆E111R group developed a fever later but were prone to show lasting intermittent fever symptoms and later onset of viremia, with longer survival times. In the low-dose SY18∆E111R group, the pigs died even though their sera were positive for antibodies. This demonstrates that antibodies are not a marker of animal resistance. The result of the deletion of the E111R gene was similar to that of the deletion of the EP296R gene, and although it had little effect, it weakened the virulence of the virus more than the deletion of the I267L gene, which had no effect at all [31,48].

In summary, *E111R* is an early-expressed gene in the ASFV genome and would encode a relatively conserved protein. Taken together, our results demonstrate that E111R is an early-expressed gene in the ASFV genome and would encode a relatively conserved protein. However, the E111R gene is a non-essential gene since its deletion from the parental virus does not significantly alter virus replication either in vitro or in vivo. Moreover, although it does involve virus virulence, the effect is negligible.

## Figures and Tables

**Figure 1 viruses-15-00890-f001:**
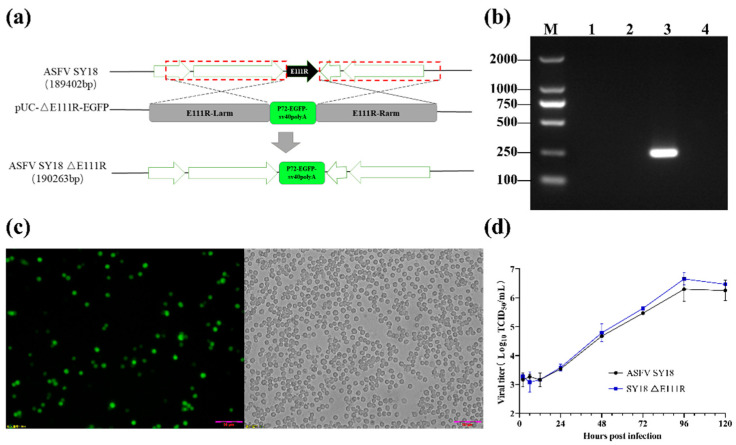
ASFV SY18ΔE111R construction and in vitro growth characteristics. (**a**) Schematic diagram of SY18ΔE111R recombinant virus construction. (**b**) The PCR amplification results of the *E111R* gene of genomic DNA. M, DL2000 DNA Marker; (1) PCR product of the *E111R* gene of the 1st-generation SY18ΔE111R strain; (2) PCR product of the *E111R* gene of the 2nd-generation SY18ΔE111R strain; (3) PCR product of the *E111R* gene of the ASFV SY18 strain; (4) negative control. (**c**) PAMs were infected by 2nd generation SY18ΔE111R after 96 h (20×, scale bar 50 μm). The 2nd generation SY18ΔE111R infected cells express green fluorescence (left), bright field plot of the 2nd generation SY18ΔE111R strain-infected cells (right). (**d**) In vitro growth characteristics of the 2nd generation SY18ΔE111R and ASFV SY18. Both viruses infected PAMs at 0.01 MOI and titrated for viral yield at 2, 6, 12, 24, 48, 72, 96, and 120 hpi; data representing the mean of the three independent experiments’ results are expressed as log10 TCID_50_/mL. Data are not significantly different for all time points (*p* > 0.05).

**Figure 2 viruses-15-00890-f002:**
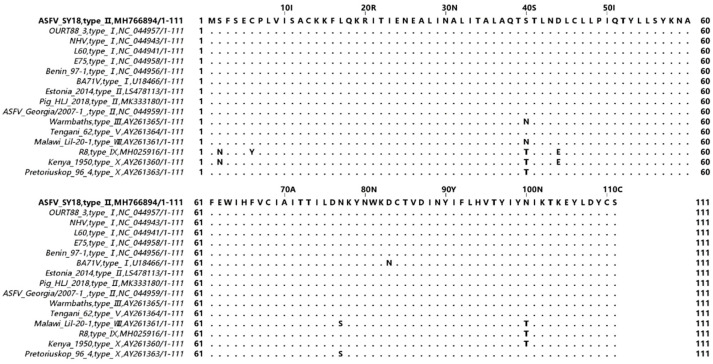
Comparison of *E111R* amino acid sequences in 16 different ASFV isolates. The *E111R* of ASFV SY18 was used as the reference sequence in the comparison, where single-letter abbreviations indicate amino acid sequences. Different letters indicate other amino acids; the same amino acid is shown as “.”. The amino acid homology of pE111R between isolates is 95.5% to 100%.

**Figure 3 viruses-15-00890-f003:**
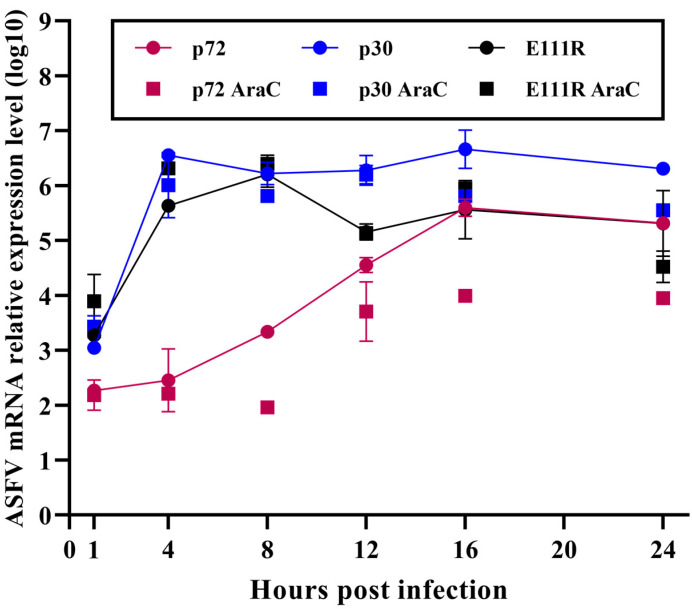
*E111R* gene transcript expression results. The graph shows the results of the expression of the *E111R*, *CP204L*, and *B646L* genes in the ASFV genome relative to the *GAPDH* gene. The RNA was collected from PAMs at 1, 4, 8, 12, 16, and 24 h after infection with ASFV (MOI = 3). Transcript levels of p30, p72, and *E111R* genes after ASFV SY18 (MOI = 3) infection of PAMs are indicated by a dash (circle marker), and data from the parallel treatment group treated with cytarabine (AraC, 100 μg/mL) are shown as squares. The Y-axis uses the decimal (log10) logarithm to indicate the relative mRNA expression levels.

**Figure 4 viruses-15-00890-f004:**
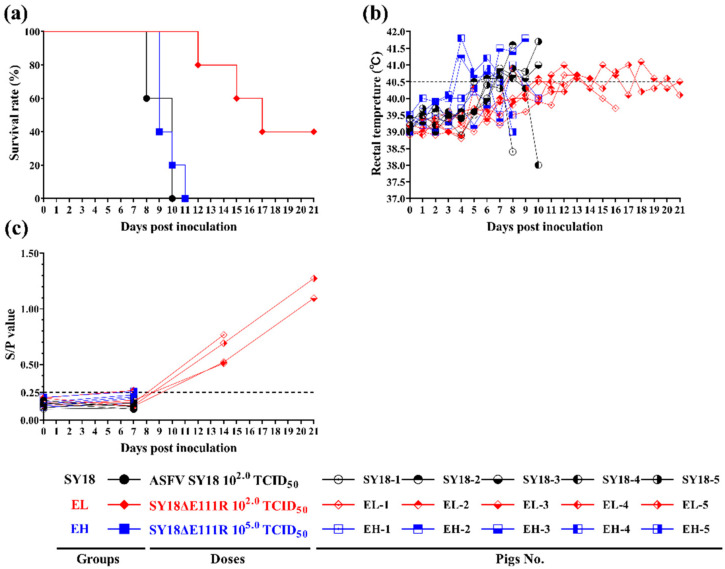
Results of survival rates, rectal temperature, and p54 antibody S/P values in pigs. (**a**) Survival rates of pigs infected with ASFV SY18 and SY18ΔE111R. (**b**) Rectal temperature data in pigs after infection. Each line represents the body temperature data of an individual animal. (**c**) The p54 antibody S/P values of animal sera. Values of S(OD_450_)/P(OD_450_) greater than 0.25 for each group were considered positive.

**Figure 5 viruses-15-00890-f005:**
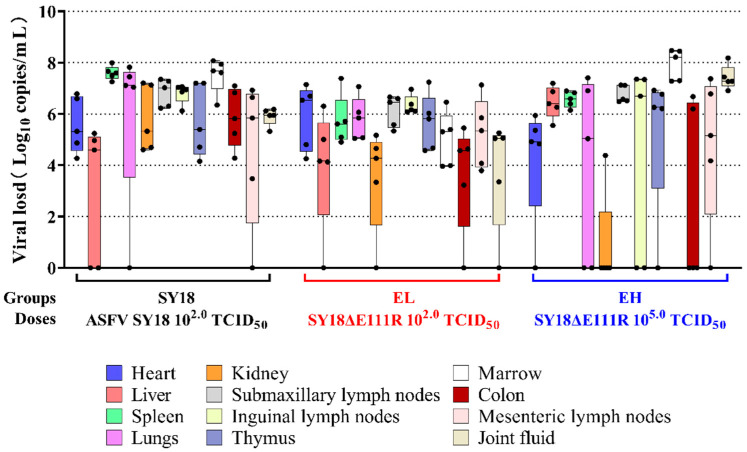
Viral load results for tissues. Black dots represent the viral load of individuals in the tissue.

**Table 1 viruses-15-00890-t001:** Primer information.

Gene	Forward Primer (5′-3′)	Reverse Primer (5′-3′)
*GAPDH*	CCTTCATTGACCTCCACTACA	GATGGCCTTTCCATTGATGAC
*B646L*	CGAACTTGTGCCAATCTC	ACAATAACCACCACGATGA
*CP204L*	TTCTTCTTGAGCCTGATGTT	TAGCGGTAGAATTGTTACGA
*E111R*	ACCAGCACGTTGAATGAT	CGTACAGTCCTTCCAGTTAT

**Table 2 viruses-15-00890-t002:** Survival and fever responses of pigs infected with SY18∆E111R and ASFV SY18.

Virus	No. of Survivors/Total	Fever (Rectal Temperature ≥ 40.5 °C)			Mean Days to Death (±SD)
		Days of Onset (±SD)	Days of Duration (±SD)	Maximum Daily Temp °C (±SD)	
ASFY SY18 (10^2.0^ TCID_50_)	0/5	6.6 (±1.0)	3.0 (±0.9)	41.18 (±0.4)	9.2 (±1.0)
SY18ΔE111R (10^2.0^ TCID_50_)	2/5	10.6 (±1.7)	4.2 (±2.2)	40.88 (±0.2)	14.67 (±2.1) ^1^
SY18ΔE111R (10^5.0^ TCID_50_)	0/5	5.8 (±1.6)	2.8 (±0.8)	41.34 (±0.4)	9.6 (±0.8)

^1^: The average time to death was only calculated for pigs that died of natural causes.

**Table 3 viruses-15-00890-t003:** Virus genome copies in blood samples; oral and anal swabs from infected pigs.

Virus	No.	ASFV Genome Copies/mL (log_10_)											
		Days Post Inoculation											
		0			7			14			21		
		Blood	Oral Swabs	Anal Swabs	Blood	Oral Swabs	Anal Swabs	Blood	Oral Swabs	Anal Swabs	Blood	Oral Swabs	Anal Swabs
ASFY SY18 (10^2.0^ TCID_50_)	SY18-1	- ^a^	-	-	6.86	4.92	4.92	/ ^b^	/	/	/	/	/
	SY18-2	-	-	-	7.86	4.45	4.80	/	/	/	/	/	/
	SY18-3	-	-	-	6.74	4.80	-	/	/	/	/	/	/
	SY18-4	-	-	-	8.34	4.63	-	/	/	/	/	/	/
	SY18-5	-	-	-	7.43	4.21	4.58	/	/	/	/	/	/
SY18ΔE111R (10^2.0^ TCID_50_)	EL-1	-	-	-	-	4.16	3.48	6.70	4.07	3.60	/	/	/
	EL-2	-	-	-	7.45	4.96	4.50	/	/	/	/	/	/
	EL-3	-	-	-	-	3.91	-	-	-	4.40	3.93	-	3.75
	EL-4	-	-	-	-	4.81	3.87	6.56	-	4.01	/	/	/
	EL-5	-	-	-	-	4.01	-	-	-	-	4.14	-	3.86
SY18ΔE111R (10^5.0^ TCID_50_)	EH-1	-	-	-	7.47	5.73	4.73	/	/	/	/	/	/
	EH-2	-	-	-	6.12	4.60	4.95	/	/	/	/	/	/
	EH-3	-	-	-	4.36	-	3.98	/	/	/	/	/	/
	EH-4	-	-	-	6.84	5.74	4.20	/	/	/	/	/	/
	EH-5	-	-	-	7.45	4.58	4.35	/	/	/	/	/	/

^a^ -: negative, ^b^ /: dead.

## Data Availability

Not applicable.

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
