# Peer review of "Evaluation of African Swine Fever Virus E111R Gene on Viral Replication and Porcine Virulence"

_viruses, 2023, doi:10.3390/v15040890_

Round 1
Reviewer 1 Report
Although studies on ASFV gene function are fundamental to increase the knowledge on ASFV virulence determinants that may have an impact on developing ASFV vaccinology, the present study is not publishable. The study presented is not based on any guiding hypothesis and this is a major drawback. It appears that authors selected the E111R by chance. Gene deletions should be identified based on hypotheses for example on their hypothetical function.
The material and method section is poorly described with the consequence of making it difficult for the reader to understand the result section. The references appear not be used appropriately. Cited papers do not seem to correlate with the sentences.
However below there are some specific comments that authors may find useful.
ABSTRACT
The abstract is well structured however, I disagree with the statement that ASFV is widespread (line 21).
INTRODUCTION
● In this section general info on virus classification is missing (i.e. virus family and genus)
● Line 52: please correct” on-going proliferation” it is not appropriate to describe the virus circulation and spread in specific geographic areas.
● The gap in knowledge the authors want to cover with the presented study is not supported by any hypothesis
MATERIAL AND METHODS
This section should be more detailed to be reproducible (i.e. info on reagents and equipment used).
● Line 91: please describe with a more appropriate term the meaning of “potent strain” maybe virulent?
● Line 92: please specify the meaning of six-generation virus” May be passaged six times? Where? In cell, pigs?
● Line 92: add a reference to the statement of 100% mortality pigs.
● Line 97: please add info on the product GM-CSF (i.e. method or reference to produce it).
● Lines 99-103: please be more specific on the methods used to verify the absence of the porcine viruses tested.
● Line 103: does porcine microvirus exist? It seems a type-error as the acronym refer to Porcine parvovirus (PPV)
● Lines 105–109: please add info on the dataset used to perform the amino-acid sequence homology of the SY18 strain
● Line 111: add info on the type of plates used
● Line 119: add reference to the method mentioned
● Line 150: add info on the monoclonal antibody to ASFV p30 protein
● Line 144: why just one MOI was tested?
● Line 152: add a reference to the Reed and Muench method
● Please add info on the type of pigs used (for example Landrance X large white)
● Add the age of pigs
● Add the method to detect the absence of other common porcine viruses
● Specify if the the TCID50 is calculated in 1 ml or 0.1ml?
● Line 160: specify where the IM injection was performed (which part of the body)
● Line 160: Any published clinical scored for ASFV were used?
● Provide detail on plasmid used or add a reference
● Justify why an home-based ELISA test was used and in case add info on its sensibility and specificity and its cut-off
● For the statistical analysis, provide details on the aim and on which data was applied. Add a reference to the Holm-Sidak test
RESULTS
The result section is very difficult to follow as the M and M section is not detailed enough.
For example in paragraph 3.3 it is not clear whether the 15-round of limited dilutions mutant corresponds to the second generation mutant. Authors should specify what they mean for the second generation. The terminology should be consistent.
In addition, it is not well described how the mutant was purified in the M and M section and this is probably why the results are difficult to read.
What mutant has been used to infect pigs? The 15-round of limited dilutions mutant?
● Which gene database was used? This info should be provided in the M and M section
● Line 231: add references
● Line 260: add details to TCID50
● Why is the parent strain administered only at high dose? We miss in this way a comparison between wild and mutant strains at low dose
● Line 274: pigs in this study were not vaccinated but infected. The aim of the study is to assess the virulence and not the potential use as a vaccine of the mutant generated. Modify the wording.
● Line 279: authors report data for specific pigs (i.e. EH-3pig) without providing this info in the M and M method. It is important to provide such detailed info or maybe is sufficient to report results stating 1 out of XXX pigs of the XY group? In the figure caption n.4 it is reported the group acronym: EL and EH that should be provided in the M and M section
● All Figure captures present a different style
DISCUSSION
The discussion section (lines 303-333) contains info that is more appropriate in an introduction section. Here authors should add info and recall data useful to discuss their results, highlighting whether they are in line or not with previous publications and how they can be considered innovative.
Line 304: are authors sure the n.24 reference is the correct one?
Line 336: authors state that the E111R is well conserved among different genotypes: Which ones? Genotype I and II only? Or others, including the attenuated strains (for example: NPH69- OURT-88)? This info should not be reported only in the paper tables. This analysis was carried out with all available ASFV sequence in Genbank?
Maybe this in silico evidence (well-conserved gene) be used as a guiding hypothesis? In other words if the E111R is conserved also in attenuated strains why it should be critical for virulence?
Reviewer 2 Report
The design of ASF virus deletion strains is currently regarded as the most promising way to develop vaccine candidates. The present work describes the development of a deletion strain based on “SY18” strain with an E111R gene deletion. It has been established that the deletion of this gene does not cause complete attenuation of the virus, but it weakens the manifestations of the disease and reduces mortality. The data presented in this paper are of considerable interest to researchers studying the functional role of the ASF virus genes or working on the development of vaccine candidates. Despite the qualitatively conducted research, the presented work requires correction of the English language.
A few notes:
Line 26 – please, add word “strain” to “SY18” (…of the lethal ASFV "SY18" strain.)
Line 52-53 – I think, this sentence should be revised. For example: The continued spread of the virus makes it increasingly difficult to prevent ASF outbreaks in countries, currently free from ASFV.
Line 63 – please, change “researched” to “studied”.
Line 64 – I’m don’t understand what author’s means under “strain genes” term. Is it species-specific genes or something else? Maybe, you just should change “strain” word to “some” or “certain”.
Line 80-81 – this sentence about homologous recombination is too hard to understand. It will be better to rewrite, or spread into two sentences.
Line 92-93 – not very good sentence. Also, please, indicate the source, from which ASFV was obtained.
Line 99 – Please, make a correction of this sentence like: “The PAMs and BMDMs were tested by PCR method for presence of ASFV, PRV etc. nucleic acid”. Also this sentence is too long, better separate it into two.
Line 122 - Table 1 - there is mistake in word “forward”.
Line 131 – Its better to change part “When the plasmid was sequenced without error” to separate sentence “Absence of nucleotide changes in recombination cassette was confirmed by sequencing”.
Line 194 – Please, specify name of gene database.
Line 236-237 – mistake in word “dilutions”.
Line 245 – please, change “ASFV genome” to “gene”.
Line 273-276 – term “vaccination” is not suitable in such context. Please, change to “infection”.
Line 307-311; 311-312 – these two sentences is too long and not very clearly articulated.
Line 322 – “longest/shortest strain” is not correct formulation. Its better to change to largest/smallest length of the genome.
Line 326 – Please, change “We still don’t know” to “It is still unknown”.
This manuscript contains interesting information obtained as a result of good work, but significant correction of the English language is needed.
Reviewer 3 Report
Several suggestions and comments for the Article "Evaluation of African swine fever virus E111R gene on viral replication and porcine virulence":
Lines 48 - to make it more clear for the readers - it is suggested to include the explanation, that Sardinia is in Italy - "and Sardinia (Italy)".
Lines 49-50 - from the description it seems that ASFV genotype II spread to Georgia, then Russia, and then to East Asia in 2018, however, since 2014 ASFV entered the eastern part of the European Union and since then spread over the European countries. A short explanatory note would provide a more clear overview of the ASFV distribution across the world.
Lines 99-100 - the authors mentioned, that "PAMs and BMDMs were tested for nucleic acid using the National Standards of the People's Republic of China...". It is suggested to make an extraction with an explanation for the readers, what are these standards, and do they differ somehow from the worldwide used standards?
Please explain, whether for the experiment ASFV of genotype II was used.
Line 114 - if abbreviations are used all over the abstract, an explanation of the abbreviations for example for "RNA", "PCR", "cDNA" should be included.
Line 269 - the authors state, that a "reduced mortality rate of 60%" was observed, however, it is not clear, how long the pigs were monitored. More detailed information on the surveillance of inoculated or intramuscularly injected pigs with different strains will provide more value for the abstract.
English language and style need to be improved due to minor spell checks are required.
Reviewer 4 Report
The manuscript by Xintao Zhou et al. describes studies on the construction and characterization of a modified strain of African swine fever virus (ASFV) that specifically lacks the E111R gene. There had been no information about the function and importance of this gene. Loss of the gene had no apparent effect on replication in cells in vitro and a modest attenuating effect within pigs (with 2 of 5 pigs surviving a low dose challenge). The authors conclude that the gene E111R is an “unsuitable option for a gene deletion vaccine” and this is essentially correct but perhaps it may be used as a second modification to further decrease the attenuation resulting from another gene deletion. This would have to be tested of course. My main concern about the data presented concerns the measurements of the mRNA expression levels by RT-qPCR, these analysis seem to lack important controls. This casts some doubt on the measurements presented.
Specific points:
1) The expression levels of 3 different ASFV genes (p72 (B646L), p30 (CP204L) and E111R) were measured by RT-qPCR. Since the mRNA sequences essentially match the genomic DNA (except for the poly(A) tail) it is important to ensure that the ASFV DNA does not contribute to these measurements of the gene expression. It appears that the authors have performed a treatment to remove genomic DNA (using gDNA wiper, see line 117) but it is not demonstrated that this treatment was successful. It seems necessary to provide the data for the results obtained without the reverse transcriptase (RT), as well as with it, to show that the results obtained with the RT actually only measure mRNA and not mRNA plus residual ASFV DNA. It is surprising to me that the changes in gene expression presented in Figure 2 only show, at most, a 2 log fold change in ASFV gene expression from 1 hr post infection through to 24 hr post infection.
2) It would be good to demonstrate that the expression of E111R is insensitive to blocking DNA replication as would be predicted for an early gene (see line 221).
3) The use of the English language needs to be significantly improved. There are many sentences in the text that do not make sense at all (e.g. lines 52-53, 165, 186, 243-245, 276-278) or are poorly written. It is important to be clear whether sequence comparisons have been performed at the amino acid level or at the DNA sequence level (see line 105). A gene has a nucleotide sequence that encodes an amino acid sequence.
4) References for certain methods are not provided (e.g. Reed & Muench method (line 152) and Holm-Sidak test (line 184)). It seems unnecessary to list all the strains used for sequence comparisons in the text (lines 201-205) when this information is present in Figure 1.
Reviewer 5 Report
The manuscript by Zhou et al, describes the evaluation of the ASFV E111R gene on viral replication and porcine virulence, which when deleted has a negligible effect on the lethality of ASFV and does not affect the ability of the virus to replicate, making E111R an unsuitable option for a gene deletion vaccine. The article here presented, challenge a very important issue, such as is the characterization of the E111R gene. The results demonstrated that when piglets were inoculated intramuscularly with a high dose of SY18ΔE111R (105 TCID50), all animals showed typical clinical signs of acute ASF, when a low dose of SY18ΔE111R (102 TCID50) is administered, it resulted in a 40% survival rate and reduced clinical signs. Therefore, authors concluded that the deletion of E111R gene has no effect on ASF SY18 strain in vitro replication and does not significantly diminish the virulence of the strain. The work is of interest to understand the role of different genes on ASFV, as it is not enough knowledge about the function of several of the viral genes. This information can be of interest and consideration for the development of attenuated vaccine candidates.
Some comments:
On the Introduction will be good to include a reference on the expansion of ASF as a pandemic, with the recent outbreak in the Dominican Republic, to keep updated with spread of the virus worldwide.
Lines 80 - 82: rewrite sentence with description on how the gene was replaced by the EGFP cassette.
Line 91: replace “potent” strain by “high virulent” strain.
Line 92: Can the authors explain what does it mean “sixth generation” virus? Is this related with number of passages of the virus in the macrophage cells? Passages may be more explicit.
Line 99: replace “nuclei acids” by “extraneous agents”.
Lines 113 – 114: Please include if cultures collected at different time points were freeze / thaw.
Line 113: replace “h” by hours post inoculation and then add abbreviation.
Line 140: authors indicate that only PCR flanking the region on the genome that has been deleted is used to determine purity of the recombinant virus, SY18ΔE111R. A full genome sequence for the entire genome will add more information to evaluate the accuracy of genetic modifications of ASF SY18ΔE111R and the integrity of the remaining virus genome.
Line 168: Indicate which techniques was used to detect ASF nucleic acid for the organs and rewrite the sentence.
Line 177: Include the times points used for the detection of p54 Abs and which ‘three serum samples” are the authors referring in here.
Figure 2 legend: Add MIO on figure legend.
Line 231: replace “Strong strain” by “highly virulent” and keep consistency through the manuscript.
Line 259: Add route of inoculation and number of pigs per group.
Line 260: be consistent on how you write the doses of virus used, add TCID50 after the name of the ASF virus used.
Lines 273 and 274: The authors are referring as “pigs vaccinated” with “SY18ΔE111R” and with “ASFV SY18”. Please replace "vaccinated" by “inoculated”, since the deleted strain is not being evaluated as a vaccine. Please keep consistency through the manuscript.
Line 284: At the end of the sentence with all tissues evaluated, refer to Table 3 and Figure 6.
Line 319: Use “less virulent” or “Attenuated” instead of “weak stains”.
It may be of interest for the authors to add as a conclusion remark that E111R gene seems to be a non-essential gene since its deletion from the wild type parental virus does not significantly alter virus replication in vitro when PAM are inoculated, or during infection in vivo.
Round 2
Reviewer 4 Report
The revised manuscript by Xintao Zhou et al., has addressed many of the comments made by the reviewers but some of the corrections/ additions still require corrections, e.g. the first sentence of the Abstract is poorly written and it should be “strain” not “stain” on line 26. I think the change to the last sentence of the Abstract is also not helpful (lines 31-33).
It should be noted that an outbreak in 2007 is in the 21st century (not 20th), see lines 47-48.
I do not think the new sentence in lines 74-75 is useful, it is too general and not well supported.
On line 96, the text refers to a -80◦C refrigerator, it should be a freezer.
On lines 119-120, the text should read: ASFV SY18 was used to infect the PAMs with a multiplicity of infection (MOI) of 3.
Line 141, the new sentence on lines 141-142 is still not correct.
Lines 146-148 still needs correction.
Lines 166-167, surely it is the antibody that is FITC-labelled not the ASFV p30 protein. This needs correction.
In Figure 2, I think the y axis should be labelled as a log10 scale.
Lines 368-369, what do the authors mean by “similar… genes”? I think the sentence needs re-writing.
In the response letter, the authors wrote: We designed to detect copy number changes in the E111R, p72 and p30 genes over 24h using a cytarabine reagent that inhibits ASFV replication or could further validate the early expression results of the E111R gene.
However, we are sorry that we were unable to give you a full response by the deadline due to a lack of key reagents. We are working hard to act and hope that you will give us some more time.
The significance of the manuscript would be increased by the demonstration that the E111R gene is an early gene. As I understand it, in the analyses by Cackett et al. (J Virol, 2020), this gene gave ambivalent results in terms of being an early or late gene. If further time is required to achieve this then that should be given in my view.
The concluding two sentences, lines 384-387, are also poor and should be rewritten.
Round 3
Reviewer 4 Report
In the revised manuscript by Xintao Zhou et al., most of the points I raised previously have been satisfactorily addressed. However, point 12 related to the identification of the E111R gene as being expressed early, i.e. before DNA replication (see lines 240-252). The authors indicated that they wished to perform a study using cytosine arabinoside (araC) as an inhibitor of DNA synthesis to strengthen their conclusion and in this latest revision they have provided data from this experiment. They used the expression of the p30 mRNA as an example of an early gene and the p72 mRNA as a late gene. The expression of E111R mRNA followed the kinetics of the p30 mRNA (see Figure 1 in the response to reviewers). However, unfortunately, the araC seems to have had no effect on the expression of the late gene (encoding p72) and this is the positive control for the effect of araC. As the authors state in their response: “the transcription of the p72, which was transcribed late in ASFV, was mainly after 10h of infection and was barely affected by cytarabine”. Thus, it is not possible to conclude anything from this experiment except that the araC did not apparently work. Did the authors test whether DNA replication was affected in the araC treated cells?
This experiment clearly needs to be repeated and the inhibitory effect of araC on viral DNA replication established. Then the results should be included in the manuscript and not just in the response to reviewers.
